# COVE: Unleashing the Diffusion Feature Correspondence for Consistent Video Editing

**Jiangshan Wang**[1][*]**, Yue Ma**[2][*]**, Jiayi Guo**[1][*]**, Yicheng Xiao**[1]**, Gao Huang**[1][†]**, Xiu Li**[1][†]
[1]Tsinghua University, [2]HKUST

https://cove-video.github.io/

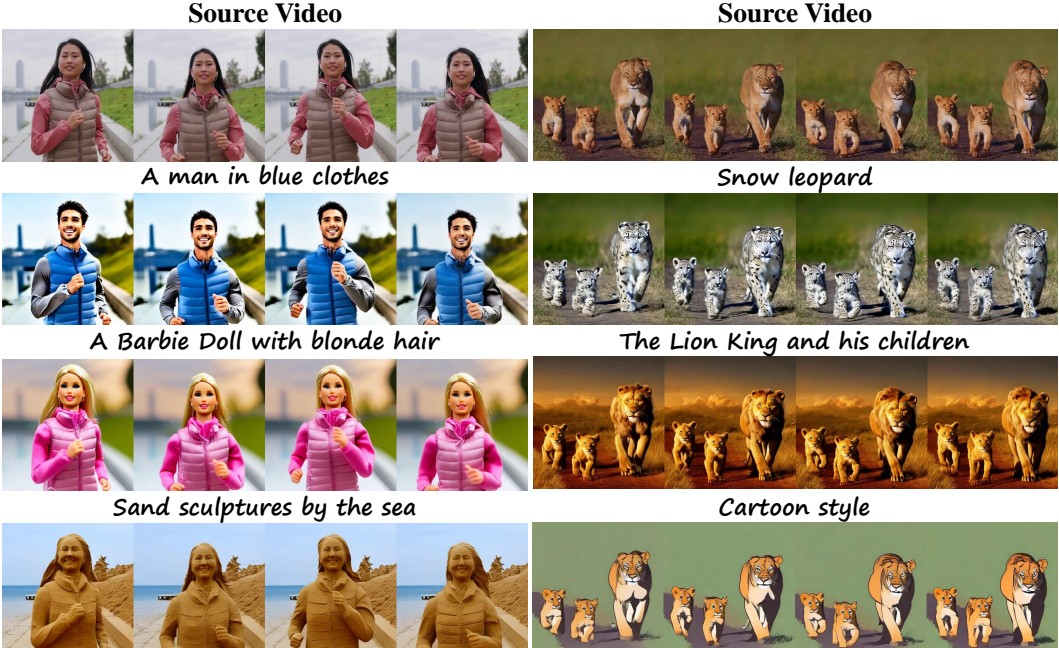

Figure 1: We propose **CO**rrespondence-guided **V**ideo **E**diting (COVE), which leverages the correspondence information of the diffusion feature to achieve consistent and high-quality video editing. Our method is capable of generating high-quality edited videos with various kinds of prompts (style, category, background, etc.) while effectively preserving temporal consistency in generated videos.

## Abstract

Video editing is an emerging task, in which most current methods adopt the pre-trained text-to-image (T2I) diffusion model to edit the source video in a zero-shot manner. Despite extensive efforts, maintaining the temporal consistency of edited videos remains challenging due to the lack of temporal constraints in the regular T2I diffusion model. To address this issue, we propose **CO**rrespondence-guided **V**ideo **E**diting (COVE), leveraging the inherent diffusion feature correspondence to achieve high-quality and consistent video editing. Specifically, we propose an efficient sliding-window-based strategy to calculate the similarity among tokens in the diffusion features of source videos, identifying the tokens with high correspondence across frames. During the inversion and denoising process, we

---

[*]Equal contribution. † Corresponding author.

38th Conference on Neural Information Processing Systems (NeurIPS 2024).

sample the tokens in noisy latent based on the correspondence and then perform self-attention within them. To save GPU memory usage and accelerate the editing process, we further introduce the temporal-dimensional token merging strategy, which can effectively reduce redundancy. COVE can be seamlessly integrated into the pre-trained T2I diffusion model without the need for extra training or optimization. Extensive experiment results demonstrate that COVE achieves the start-of-the-art performance in various video editing scenarios, outperforming existing methods both quantitatively and qualitatively. The code will be release at https://github.com/wangjiangshan0725/COVE

# 1   Introduction

Diffusion models [27, 63, 65] have shown exceptional performance in image generation [57], thereby inspiring their application in the field of image editing [6, 25, 7, 53, 67, 24]. These approaches typically leverage a pre-trained Text-to-Image (T2I) stable diffusion model [57], using DDIM [64] inversion to transform source images into noise, which is then progressively denoised under the guidance of a prompt to generate the edited image.

Despite satisfactory performance in image editing, achieving high-quality video editing remains challenging. Specifically, unlike the well-established open-source T2I stable diffusion models [57], comparable T2V diffusion models are not as mature due to the difficulty of modeling complicated temporal motions, and training a T2V model from scratch demands substantial computational resources [26, 29, 62]. Consequently, there is a growing focus on adapting the pre-trained T2I diffusion for video editing [16, 33, 11, 72, 73, 54]. In this case, maintaining temporal consistency in edited videos is one of the biggest challenges, which requires the generated frames to be stylistically coherent and exhibit smooth temporal transitions, rather than appearing as a series of independent images. Numerous methods have been working on this topic while still facing various limitations, such as the inability to ensure fine-grained temporal consistency (leading to flickering [33, 54] or blurring [16] in generated videos), requiring additional components [30, 72, 11, 73] or needing extra training or optimization [73, 69, 41], etc.

In this work, our goal is to achieve highly consistent video editing by leveraging the intra-frame correspondence relationship among tokens, which is intuitively closely related to the temporal consistency of videos: If corresponding tokens across frames exhibit high similarity, the resulting video will thus demonstrate high temporal consistency. Taking a video of a man as an example, if the token representing his nose has high similarity across frames, his nose will be unlikely to deform or flicker throughout the video. However, how to obtain accurate correspondence information among tokens is still largely under-explored in existing works, although the intrinsic characteristic of the video editing task (i.e., the source video and edited

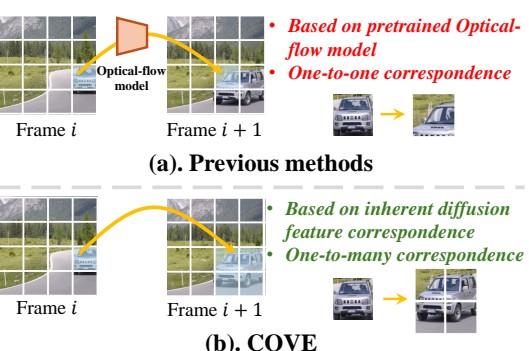

(a). Previous methods

(b). COVE

Figure 2: **Comparison between COVE (our method) and previous methods**[11, 73].

video are expected to share similar motion and semantic layout) determines that it naturally exists in the source video. Some previous methods [11, 73] leverage a pre-trained optical-flow model to obtain the flowing trajectory of each token across frames, which can be seen as a kind of coarse correspondence information. Despite the self-attention among tokens in the same trajectory can enhance the temporal consistency of the edited video, it still encounters two primary limitations: Firstly, these methods heavily rely on a highly accurate pre-trained optical-flow model to obtain the correspondence relationship of tokens, which is not available in many scenarios [32]. Secondly, supposing we have access to an extremely accurate optical-flow model, it is still only able to obtain the coarse one-to-one correspondence among tokens in different frames (Figure 2a), which would lead to the loss of information because one token is highly likely to correspond to multiple tokens in other frames in most cases (Figure 2b).

Addressing these problems, we notice that the inherent diffusion features naturally contain precise correspondence information. For instance, it is easy to find the corresponding points between two images by extracting their diffusion features and calculating the cosine similarity between tokens [66]. However, until now none of the existing works have successfully utilized this characteristic in more complicated and challenging tasks such as video editing. In this paper, we propose COVE, which is the first work unleashing the potential of inherent diffusion feature correspondence to significantly enhance the quality and temporal consistency in video editing. Given a source video, we first extract the diffusion feature of each frame. Then for each token in the diffusion feature, we obtain its corresponding tokens in other frames based on their similarity. Within this process, we propose a sliding-window-based approach to ensure computational efficiency. In our sliding-window-based method, for each token, it is only required to calculate the similarity between it and the tokens in the next frame located within a small window, identifying the tokens with the top $K$ ($K > 1$) highest similarity. After the correspondence calculation process, for each token, the coordinates of its $K$ corresponding tokens in each other frame can be obtained. During the inversion and denoising process, we sample the tokens in noisy latents based on the obtained coordinates. To reduce the redundancy and accelerate the editing process, token merging is applied in the temporal dimension, which is followed by self-attention. Our method can be seamlessly integrated into the off-the-shelf T2I diffusion model without extra training or optimization. Extensive experiments demonstrate that COVE significantly improves both the quality and the temporal consistency of generated videos, outperforming a wide range of existing methods and achieving state-of-the-art results.

## 2 Related Works

### 2.1 Diffusion-based Image and Video Generation.

Diffusion Models [27, 63, 65] have recently showcased impressive results in image generation, which generates the image through gradual denoising from the standard Gaussian noise[12, 13, 52, 21, 57, 64, 22]. A large number of efforts on diffusion models [28, 34, 60] has enabled it to be applied to numerous scenarios [3, 15, 35, 39, 44, 49, 50, 14, 58, 9, 23, 47, 42, 18]. With the aid of large-scale pretraining [55, 61], text-to-image diffusion models exhibit remarkable progress in generating diverse and high-quality images [51, 71, 56, 57, 59, 19, 48, 18]. ControlNet [75] enables users to provide structure or layout information for precise generation. Naturally, diffusion models have found application in video synthesis, often by integrating temporal layers into image-based DMs [4, 26, 29, 70, 10]. Despite successes in unconditional video generation [29, 74, 45], text-to-video diffusion models lag behind their image counterparts.

### 2.2 Text-to-Video Editing.

There are increasing works adopting the pre-trained text-to-image diffusion model to the video editing task [43, 68, 69, 46, 20], where keeping the temporal consistency in the generated video is the most challenging. Recently, a large number of works focusing on zero-shot video editing has been proposed. FateZero [54] proposes to use attention blending to achieve high-quality edited videos while struggling to edit long videos. TokenFlow [16] reduces the effects of flickering through the linear combinations between diffusion features, while the smoothing strategy can cause blurring in the generated video. RAVE [33] proposes the randomized noise shuffling method, suffering the problem of fine details flickering. There are also a large number of methods that enhance the temporal consistency with the aid of pre-trained optical-flow models [73, 72, 11, 30]. Although the effectiveness of them, all of them severely rely on a pre-trained optical-flow model. Recent works [66] illustrate that the diffusion feature contains rich correspondence information. Although VideoSwap [17] adopts this characteristic by tracking the key points across frames, it still needs users to provide the key points as the extra addition manually.

## 3 Method

In this section, we will introduce COVE in detail, which can be seamlessly integrated into the pre-trained T2I diffusion model for high-quality and consistent video editing without the need for training or optimization (Figure 3). Specifically, given a source video, we first extract the diffusion feature of each frame using the pre-trained T2I diffusion model. Then, we calculate the one-to-many

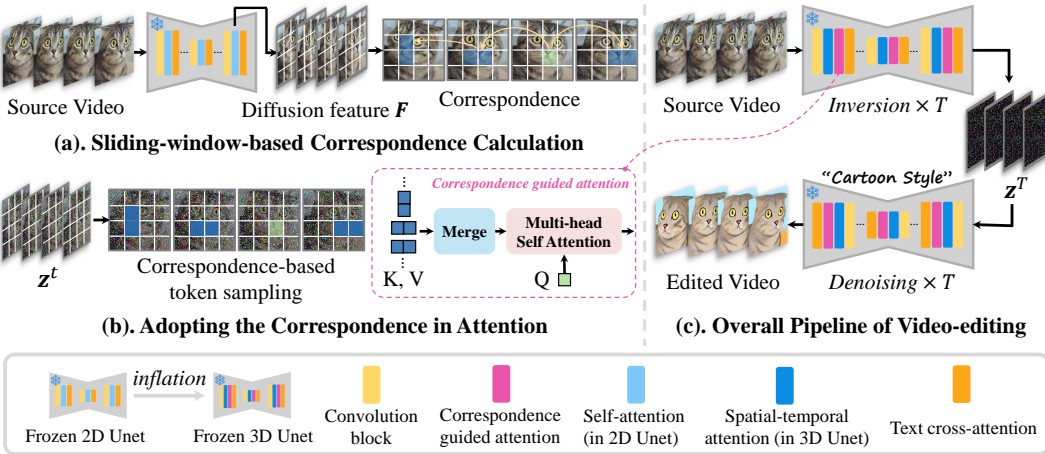

**(a). Sliding-window-based Correspondence Calculation**

**(b). Adopting the Correspondence in Attention**

**(c). Overall Pipeline of Video-editing**

Frozen 2D Unet — inflation → Frozen 3D Unet | Convolution block | Correspondence guided attention | Self-attention (in 2D Unet) | Spatial-temporal attention (in 3D Unet) | Text cross-attention

Figure 3: **The overview of COVE.** (a). Given a source video, we extract the diffusion feature of each frame using the pre-trained T2I model and calculate the correspondence among tokens (detailed in Figure 4). (b). During the video editing process, we sample the tokens in noisy latent based on correspondence and apply self-attention among them. (c). The correspondence-guided attention can be seamlessly integrated into the T2I diffusion model for consistent and high-quality video editing.

correspondence of each token across frames based on cosine similarity (Figure 3a). To reduce resource consumption during correspondence calculation, we further introduce an efficient sliding-window-based strategy (Figure 4). During each timestep of inversion and denoising in video editing, the tokens in the noisy latent are sampled based on the correspondence and then merged. Through the self-attention among merged tokens (Figure 3b), the quality and temporal consistency of edited videos are significantly enhanced.

## 3.1 Preliminary

**Diffusion Models.** DDPM [27] is the latent generative model trained to reconstruct a fixed forward Markov chain $x_1, \ldots, x_T$. Given the data distribution $x_0 \sim q(x_0)$, the Markov transition $q(x_t|x_{t-1})$ is defined as a Gaussian distribution with a variance schedule $\beta_t \in (0, 1)$.

$$q(\boldsymbol{x}_t|\boldsymbol{x}_{t-1}) = \mathcal{N}(\boldsymbol{x}_t; \sqrt{1-\beta_t}\boldsymbol{x}_{t-1}, \beta_t\mathbf{I}). \tag{1}$$

To generate the Markov chain $x_0, \cdots, x_T$, DDPM leverages the reverse process with a prior distribution $p(x_T) = \mathcal{N}(x_T; 0, \mathbb{I})$ and Gaussian transitions. A neural network $\epsilon_\theta$ is trained to predict noises, ensuring that the reverse process is close to the forward process.

$$p_\theta(\boldsymbol{x}_{t-1}|\boldsymbol{x}_t) = \mathcal{N}(\boldsymbol{x}_{t-1}; \mu_\theta(\boldsymbol{x}_t, \boldsymbol{\tau}, t), \Sigma_\theta(\boldsymbol{x}_t, \boldsymbol{\tau}, t)), \tag{2}$$

where $\boldsymbol{\tau}$ indicates the textual prompt. $\mu_\theta$ and $\Sigma_\theta$ are predicted by the denoising model $\epsilon_\theta$. Since the diffusion and denoising process in the pixel space is computationally extensive, latent diffusion [57] is proposed to address this issue by performing these processes in the latent space of a VAE [37].

**DDIM Inversion.** DDIM can convert random noise to a deterministic $\boldsymbol{x}_0$ during sampling [64, 13]. The inversion process in deterministic DDIM can be formulated as follows:

$$\boldsymbol{x}_{t+1} = \sqrt{\frac{\alpha_{t+1}}{\alpha_t}}\boldsymbol{x}_t + \sqrt{\alpha_{t+1}}\left(\sqrt{\frac{1}{\alpha_{t+1}-1}} - \sqrt{\frac{1}{\alpha_t}-1}\right)\epsilon_\theta(\boldsymbol{x}_t), \tag{3}$$

where $\alpha_t$ denotes $\prod_{i=1}^t(1-\beta_i)$. The inversion process of DDIM is utilized to transform the input $\boldsymbol{x}_0$ into $\boldsymbol{x}_T$, facilitating subsequent tasks such as reconstruction and editing.

## 3.2 Correspondence Acquisition

As discussed in Section 1, intra-frame correspondence is crucial for the quality and temporal consistency of edited videos while remaining largely under-explored in existing works. In this section, we introduce our method for obtaining correspondence relationships among tokens across frames.

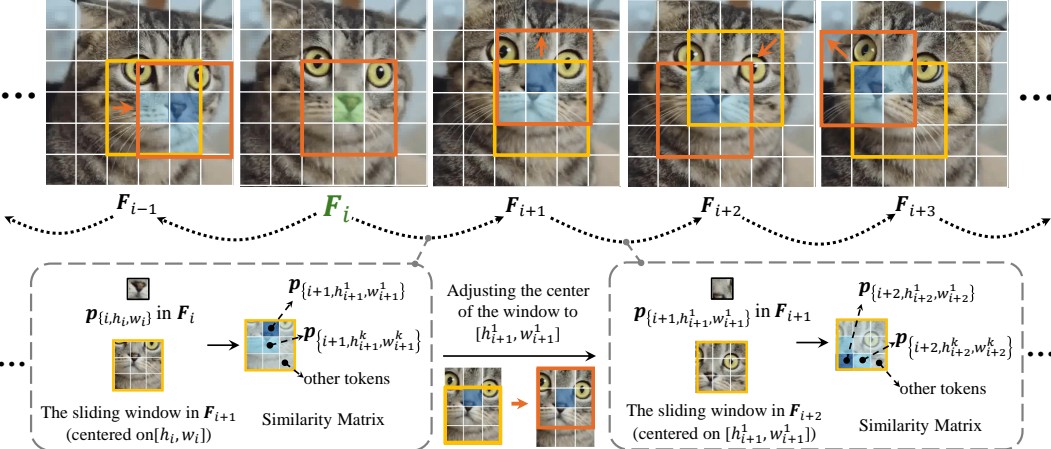

Figure 4: **Sliding-window-based strategy for correspondence calculation.** ▇ represents the token $\boldsymbol{p}_{\{i,h_i,w_i\}}$. ▇ and ▇ represents the obtained corresponded tokens in other frames.

**Diffusion Feature Extraction.** Given a source video $\boldsymbol{V}$ with $N$ frames, a VAE [37] is employed on each frame to extract the latent features $\boldsymbol{Z} = \{\boldsymbol{z}_1, \cdots, \boldsymbol{z}_N\}$, where $\boldsymbol{Z} \in \mathbb{R}^{N \times H \times W \times d}$. Here, $H$ and $W$ denote the height and width of the latent feature and $d$ denotes the dimension of each token. For each frame of $\boldsymbol{Z}$, we add noise of a specific timestep $t$ and feed the noisy frame $\boldsymbol{Z}^t = \{\boldsymbol{z}_1^t, \cdots, \boldsymbol{z}_N^t\}$ into the pre-trained T2I model $f_\theta$ respectively. The diffusion feature (i.e., the intermediate feature from the U-Net decoder) is extracted through a single step of denoising [66]:

$$\boldsymbol{F} = \{\boldsymbol{F}_i\} = \{f_\theta(\boldsymbol{z}_i^t)\}, i \in \{1, \cdots, N\}, \tag{4}$$

where $\boldsymbol{F} \in \mathbb{R}^{N \times H \times W \times d}$, denoting the normalized diffusion feature of each frame.

**One-to-many Correspondence Calculation.** For each token within the diffusion feature $\boldsymbol{F}$, its corresponding tokens in other frames are identified based on the cosine similarity. Without loss of generality, we could consider a specific token $\boldsymbol{p}_{\{i,h_i,w_i\}}$ in the $i$th frame $\boldsymbol{F}_i$ with the coordinate $[h_i, w_i]$. Unlike previous methods where only one corresponding token of $\boldsymbol{p}_{\{i,h_i,w_i\}}$ can be identified in each frame (Figure 2a), our method can obtain the one-to-many correspondences simply by selecting tokens with the top $K$ highest similarity in each frame. We record their coordinates, which are used for sampling the tokens for self-attention in the subsequent inversion and denoising process. To implement this process, the most straightforward method is through a direct matrix multiplication of the normalized diffusion feature $\boldsymbol{F}$.

$$\boldsymbol{S} = \boldsymbol{F} \cdot \boldsymbol{F}^T, \tag{5}$$

where $\boldsymbol{S} \in \mathbb{R}^{(N \times H \times W) \times (N \times H \times W)}$ represents the cosine similarity between each token and all tokens in the diffusion feature of the video.

The similarity between $\boldsymbol{p}_{\{i,h_i,w_i\}}$ and all $N \times H \times W$ tokens in the feature is given by $\boldsymbol{S}[i, h_i, w_i, :, :, :]$. The coordinates of the corresponding tokens in the $j$th frame ($j \in \{1, \cdots, N\}$) are then obtained by selecting the tokens with the top $K$ similarities in the $j$th frame.

$$h_j^k, w_j^k = \text{top-}k\text{-argmax}_{(x^k, y^k)}(\boldsymbol{S}[i, h_i, w_i, j, x^k, y^k]), \tag{6}$$

Here the top-$k$-argmax$(\cdot)$ denotes the operation to find coordinates of the top $K$ biggest values in a matrix, where $k \in \{1, \cdots, K\}$. $[h_j^k, w_j^k]$ represents the coordinates of the token in $j$th frame which has highest similarity with $\boldsymbol{p}_{\{i,h_i,w_i\}}$. A similar process can be conducted for each token of $\boldsymbol{F}$, thereby obtaining their correspondences among frames.

**Sliding-window Strategy.** Although the one-to-many correspondence among tokens can be effectively obtained through the above process, it requires excessive computational resources because $(N \times H \times W)$ is always a huge number, especially in long videos. As a result, the computational complexity of this process is extremely high, which can be represented as $\mathcal{O}(N^2 \times H^2 \times W^2 \times d)$. At the same time, multiplication between these two huge matrices consumes a substantial amount

of GPU memory in practice. These limitations severely limit its applicability in many real-world scenarios, such as on mobile devices.

To address the above problem, we further propose the sliding-window-based strategy as an alternative, which not only effectively obtains the one-to-many correspondences but also significantly reduces the computational overhead (Figure 4). Firstly, for the token $\boldsymbol{p}_{\{i,h_i,w_i\}}$, it is only necessary to calculate its similarity with the tokens in the next frame $\boldsymbol{F}_{i+1}$ instead of in all frames, i.e.,

$$\boldsymbol{S}_i = \boldsymbol{F}_i \cdot \boldsymbol{F}_{i+1}^T. \tag{7}$$

$\boldsymbol{S}_i \in \mathbb{R}^{H \times W \times H \times W}$ denotes the similarity between the tokens in $i$th frame and those in $(i+1)$th frame. The overall similarity matrix is $\boldsymbol{S} = \{\boldsymbol{S}_i\}, i \in \{1, 2, \cdots, N-1\}$, where $\boldsymbol{S} \in \mathbb{R}^{(N-1) \times H \times W \times H \times W}$. Then, we obtain the $K$ corresponded tokens of $\boldsymbol{p}_{\{i,h_i,w_i\}}$ in $\boldsymbol{F}_{i+1}$ through $\boldsymbol{S}_i$,

$$h_{i+1}^k, w_{i+1}^k = \text{top-}k\text{-argmax}_{(x^k,y^k)}(\boldsymbol{S}_i[h_i, w_i, x^k, y^k]), \tag{8}$$

For tokens in $(i+2)$th frame, instead of considering $\boldsymbol{p}_{\{i,h_i,w_i\}}$, we identify the tokens in $(i+2)$th frame which have the top $K$ largest similarity with the token $\boldsymbol{p}_{\{i+1,h_{i+1}^1,w_{i+1}^1\}}$ through the $\boldsymbol{S}_{i+1}$. Similarly, we can obtain the corresponding token in other future or previous frames.

$$h_{i+2}^k, w_{i+2}^k = \text{top-}k\text{-argmax}_{(x^k,y^k)}(\boldsymbol{S}_{i+1}[h_{i+1}^1, w_{i+1}^1, x^k, y^k]), \tag{9}$$

Through the above process, the overall complexity is reduced to $\mathcal{O}((N-1) \times H^2 \times W^2 \times d)$. Furthermore, it is noteworthy that frames in a video exhibit temporal continuity, implying that the spatial positions of corresponding tokens are unlikely to change significantly between consecutive frames. Consequently, for the token $\boldsymbol{p}_{\{i,h_i,w_i\}}$, it is enough to only calculate the similarity within a small window of length $l$ in the adjacent frame, where $l$ is much smaller than $H$ and $W$,

$$\boldsymbol{F}_{i+1}^w = \boldsymbol{F}_{i+1}[h_i - l/2 : h_i + l/2, w_i - l/2 : w_i + l/2, :]. \tag{10}$$

$\boldsymbol{F}_{i+1}^w \in \mathbb{R}^{l \times l \times d}$ represents the tokens in $\boldsymbol{F}_{i+1}$ within the sliding window. We calculate the cosine similarity between $\boldsymbol{p}_{\{i,h_i,w_i\}}$ and the tokens in $\boldsymbol{F}_{i+1}^w$, selecting tokens with top $K$ highest similarity within $\boldsymbol{F}_{i+1}^w$. This approach further reduces the computational complexity to $\mathcal{O}((N-1) \times H \times W \times l^2 \times d)$ and the GPU memory consumption is also significantly reduced in practice. Additionally, it is worth noting that calculating correspondence information from the source video is only conducted once before the inversion and denoising process of video editing. Compared with the subsequent editing process, this process only takes negligible time.

### 3.3 Correspondence-guided Video Editing.

In this section, we explain how to apply the correspondence information to the video editing process (Figure 3c). In the inversion and denoising process of video editing, we sample the corresponding tokens from the noisy latent for each token based on the coordinates obtained in Section 3.2. For the token $\boldsymbol{z}_{i,h_i,w_i}^t$, the set of corresponding tokens in other frames at a timestep $t$ is:

$$\boldsymbol{Corr} = \{\boldsymbol{z}_{\{j,h_j^k,w_j^k\}}^t\}, j \in \{1, \cdots, i-1, i+1, \cdots, N\}, k \in \{1, \cdots, K\}. \tag{11}$$

We merge these tokens following [5], which can accelerate the editing process and reduce GPU memory usage without compromising the quality of editing results:

$$\widetilde{\boldsymbol{Corr}} = \text{Merge}(\boldsymbol{Corr}). \tag{12}$$

Then, the self-attention is conducted on the merged tokens,

$$\boldsymbol{Q} = \boldsymbol{z}_{\{i,h_i,w_i\}}^t, \boldsymbol{K} = \boldsymbol{V} = \widetilde{\boldsymbol{Corr}}, \tag{13}$$

$$\text{Attention}(\boldsymbol{Q}, \boldsymbol{K}, \boldsymbol{V}) = \text{SoftMax}\left(\frac{\boldsymbol{Q} \cdot \boldsymbol{K}^T}{\sqrt{d_k}}\right) \cdot \boldsymbol{V}, \tag{14}$$

where $\sqrt{d_k}$ is the scale factor. The above process of correspondence-guided attention is illustrated in Figure 3b. Following the previous methods [73, 11], we also retain the spatial-temporal attention [69] in the U-Net. In spatial-temporal attention, considering a query token, all tokens in the video serve as keys and values, regardless of their relevance to the query. This correspondence-agnostic self-attention is not enough to maintain temporal consistency, introducing irrelevant information into each token, and thus causing serious flickering effects [11, 16]. Our correspondence-guided attention can significantly alleviate the problems of spatial-temporal attention, increasing the similarity of corresponding tokens and thus enhancing the temporal consistency of the edited video.

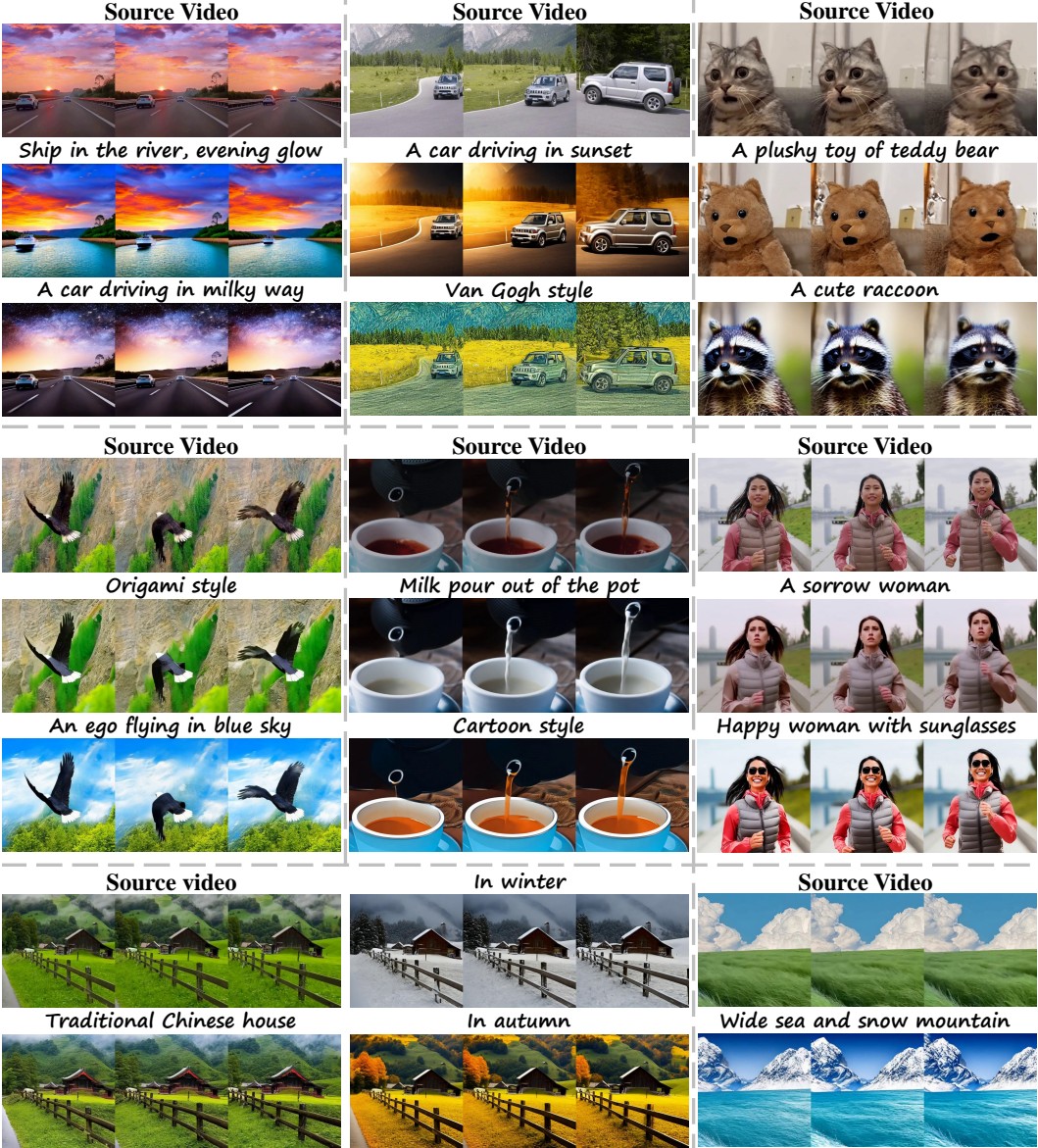

Figure 5: **Qualitative results of COVE.** COVE can effectively handle various types of prompts, generating high-quality videos. For both global editing (e.g., style transferring and background editing) and local editing (e.g., modifying the appearance of the subject), COVE demonstrates outstanding performance. Results are best-viewed zoomed-in.

## 4 Experiment

### 4.1 Experimental Setup

In the experiment, we adopt Stable Diffusion (SD) 2.1 from the official Huggingface repository for COVE, employing 100 steps of DDIM inversion and 50 steps of denoising. To extract the diffusion feature, the noise of the specific timestep $t = 261$ is added to each frame of the source video following [66]. The feature is then extracted from the intermediate layer of the 2D Unet decoder during a single step of denoising. The window size $l$ is set to 9 for correspondence calculation, and $k$ is set to 3 for correspondence-guided attention. The merge ratio for token merging is 50%. For both qualitative and quantitative evaluation, we select 23 videos from social media platforms such as TikTok and other publicly available sources [1, 2]. Among these 23 videos, 3 videos have a length of 10 frames, 15 videos have a length of 20 frames, and 5 videos have a length of 32 frames. The experiments are

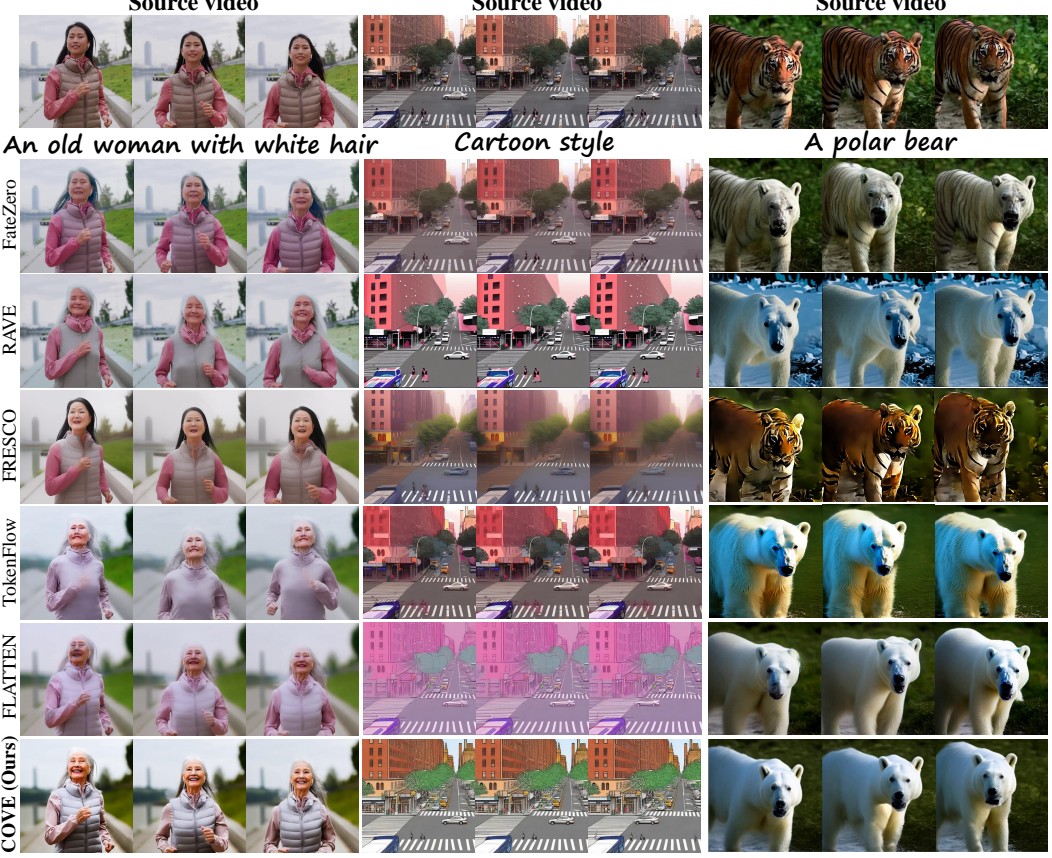

Figure 6: **Qualitative comparison of COVE and various state-of-the-art methods.** Our method outperforms previous methods across a wide range of source videos and editing prompts, demonstrating superior visual quality and temporal consistency. Results are best-viewed zoomed-in.

conducted on a single RTX 3090 GPU for our method unless otherwise specified. We compare COVE with 5 baseline methods: FateZero [54], TokenFlow [16], FLATTEN [11], FRESCO [73] and RAVE [33]. For all of these baseline methods, we follow the default settings from their official Github repositories. The more detailed experimental settings of our method are provided in Appendix A.

## 4.2 Qualitative Results

We evaluate COVE on various videos under different types of prompts including both global and local editing (Figure 5). Global editing mainly involves background editing and style transferring. For background editing, COVE can modify the background while keeping the subject of the video unchanged (e.g. Third row, first column. "`a car driving in milky way`"). For style transfer, COVE can effectively modify the global style of the source video according to the prompt (e.g. Third row, second column. "`Van Gogh style`"). Our prompts for local editing include changing the subject of the video to another one (e.g. Third row, third column. "`A cute raccoon`") and making local edits to the subject (e.g. fifth row, third column. "`A sorrow woman`"). For all of these editing tasks, COVE demonstrates outstanding performance, generating frames with high visual quality while successfully preserving temporal consistency. We also compare COVE with a wide range of state-of-the-art video editing methods (Figure 6). The experimental results illustrate that COVE effectively edits the video with high quality, significantly outperforming the previous methods.

## 4.3 Quantitative Results

For quantitative comparison, we follow the metrics proposed in VBench [31], including Subject Consistency, Motion Smoothness, Aesthetic Quality, and Imaging Quality. Among them, Subject Consistency assesses whether the subject (e.g., a person) remains consistent throughout the whole

video by calculating the similarity of DINO [8] feature across frames. Motion Smoothness utilizes the motion priors of the video frame interpolation model [40] to evaluate the smoothness of the motion in the generated video. Aesthetic Quality uses the LAION aesthetic predictor [38] to assess the artistic and beauty value perceived by humans on each frame. Imaging Quality evaluates the degree of distortion in the generated frames (e.g., blurring, flickering) through the MUSIQ [36] image quality predictor. Each video undergoes editing with 3 global prompts (such as style transferring, background editing, etc.) and 2 local prompts (such as editing the appearance of the subject in the video), generating a total of 115 text-video pairs. For each metric, we report the average score of these 115 videos. We further conducted a user study with 45 participants following [73]. Participants are required to choose the most preferable results among these methods. The result is shown in Table 1. Among various methods, COVE achieves outstanding performance in both qualitative metrics and user studies, further demonstrating its superiority.

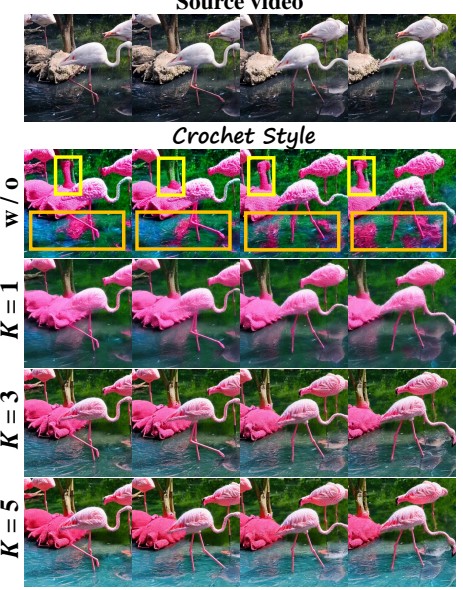

Source video

Crochet Style

|  | Subject Consistency | Motion Smoothness | Aesthetic Quality | Imaging Quality | User Study |
|---|---|---|---|---|---|
| FateZero [54] | 0.9622 | 0.9547 | 0.6258 | 0.6951 | 7.4% |
| TokenFlow [16] | 0.9513 | 0.9803 | 0.6904 | 0.7354 | 13.0% |
| FLATTEN [11] | 0.9617 | 0.9622 | 0.6544 | 0.7155 | 14.8% |
| FRESCO [73] | 0.9358 | 0.9737 | 0.6582 | 0.6331 | 9.2% |
| RAVE [33] | 0.9518 | 0.9732 | 0.6369 | 0.7355 | 11.1% |
| **COVE (Ours)** | **0.9731** | **0.9892** | **0.7122** | **0.7441** | **44.5%** |

Table 1: **Quantitative comparison** among COVE and a wide range of state-of-the-art video editing methods. The evaluation metrics[31] can effectively reflect the temporal consistency and frame quality of generated videos. COVE illustrates superior performance in both keeping the temporal consistency and generating frames with high quality in edited videos.

|  | Subject Consistency | Motion Smoothness | Aesthetic Quality | Imaging Quality |
|---|---|---|---|---|
| w/o | 0.9431 | 0.9049 | 0.6913 | 0.7132 |
| $K = 1$ | 0.9637 | 0.9817 | 0.6979 | 0.7148 |
| $K = 3$ | 0.9731 | **0.9892** | 0.7122 | **0.7441** |
| $K = 5$ | **0.9745** | 0.9886 | **0.7167** | 0.7429 |

Table 2: **Ablation study on the value of $K$ in correspondence-guided attention.** w / o means without correspondence-guided attention in Unet. When $K = 3$ the quality of the video is the best.

Figure 7: Ablation study about the correspondence-guided attention and the value of $K$. w / o means do not apply correspondence-guided attention.

## 4.4 Ablation Study

We conduct an ablation study to illustrate the effectiveness of the **Correspondence-guided attention** and the number of tokens selected in each frame (i.e., the value of $K$). The experimental results (Table 2 and Figure 7) illustrate that without correspondence-guided attention, the edited video exhibits obvious temporal inconsistency and flickering effects (which is marked in yellow and orange boxes in Figure 7), thus severely impairing the visual quality. As $K$ increases from 1 to 3, the generated video contains more fine-grained details, exhibiting better visual quality. However, further increasing $K$ to 5 does not significantly improve the video quality. We also illustrate the effectiveness of **temporal dimensional token merging**. By merging the tokens with high correspondence across frames, the editing process becomes more efficient (Table 3) while there is no significant decrease in the quality of the edited video (Figure 8). The ablation of the **sliding-window size** $l$ is shown in Appendix B. If the window size is too small, the actual corresponding token may not be included within the window, resulting in suboptimal correspondence and poor editing results. On the other hand, a too-large window size is not necessary for identifying the corresponding tokens, which would lead to high computational complexity and excessive memory usage. The experiment results illustrate that $l = 9$ is suitable to strike a balance. Additionally, we also **visualize the correspondence** obtained by COVE, which is shown in Appendix C.

| Correspondence Guided Attention | Token Merging | Speed | Memory Usage |
|---|---|---|---|
| ✗ | ✗ | 2.2 min | 9 GB |
| ✓ | ✗ | 2.7 min | 14 GB |
| ✓ | ✓ | 2.4 min | 11 GB |

Table 3: **Ablation Study on the effect of temporal dimensional token merging.** Temporal dimensional token merging can speed up the editing process and save GPU memory usage while hardly impairing the quality of the generated video. The experiment is conducted on a single RTX3090 GPU with a 10-frame source video. $k$ is set to 3.



Figure 8: Token merging would not impair the quality of edited video.

## 5    Conclusion

In this paper, we propose COVE, which is the first to explore how to employ inherent diffusion feature correspondence in video editing to enhance editing quality and temporal consistency. Through the proposed efficient sliding-window-based strategy, the one-to-many correspondence relationship among tokens across frames is obtained. During the inversion and denoising process, self-attention is performed within the corresponding tokens to enhance temporal consistency. Additionally, we also apply token merging in the temporal dimension to improve the efficiency of the editing process. Both quantitative and qualitative experimental results demonstrate the effectiveness of our method, which outperforms a wide range of previous methods, achieving state-of-the-art editing quality.

**Limitaions.** The limitation of our method is discussed in Appendix G.

## Acknowledgments and Disclosure of Funding

This work was supported by the STI 2030-Major Projects under Grant 2021ZD0201404.

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

# Appendix

## A   Detailed Experimental Settings

In the experiment, the size of all source videos is $512 \times 512$. We adopt Stable Diffusion (SD) 2.1 from the official Huggingface repository for our method. To extract the diffusion feature, following [66], the noise of the timestep $t = 261$ is added to each frame of the source video. The noisy frames of video are fed into the U-net, the feature is extracted from the intermediate layer of the 2D Unet decoder. The height and weight of the diffusion feature is 64. Following previous works, at the first 40 timesteps, the diffusion features are saved during DDIM inversion and are further injected during denoising. For Spatial-temporal attention, we use the xFormers to reduce memory consumption, while it is not used in correspondence-guided attention.

## B   Ablation Study on the Window Size

To illustrate the influence of window size $l$, we conduct the experiment on a video with 20 frames on a single A100 GPU. During the correspondence calculation process, we calculate the theoretical computational complexity, which is the total number of multiplications and additions required. We also record actual GPU memory consumed under different window sizes, the result is shown in Appendix B. With our sliding window strategy, the computational complexity and the GPU memory in the correspondence calculation process are significantly reduced. The visualization result is shown in Fig. 9. If the window size is too small, the motion in the video cannot be tracked, causing unsatisfying results. We choose $l = 9$ for the experiments in other sections, which can achieve a balance between the memory consumed and the quality of the edited video.

| Window Size ($l$) | 3 | 9 | 15 | w/o |
|---|---|---|---|---|
| Computational Complexity ($\times 10^9$) | 0.448 | 4.03 | 11.2 | 241 |
| GPU Memory (GB) | 11 | 14 | 18 | 32 |

Table 4: **Ablation Study on the window size** $l$. w/o means that the sliding-window strategy is not applied. The sliding window strategy can significantly reduce the use of computational complexity and GPU memory.

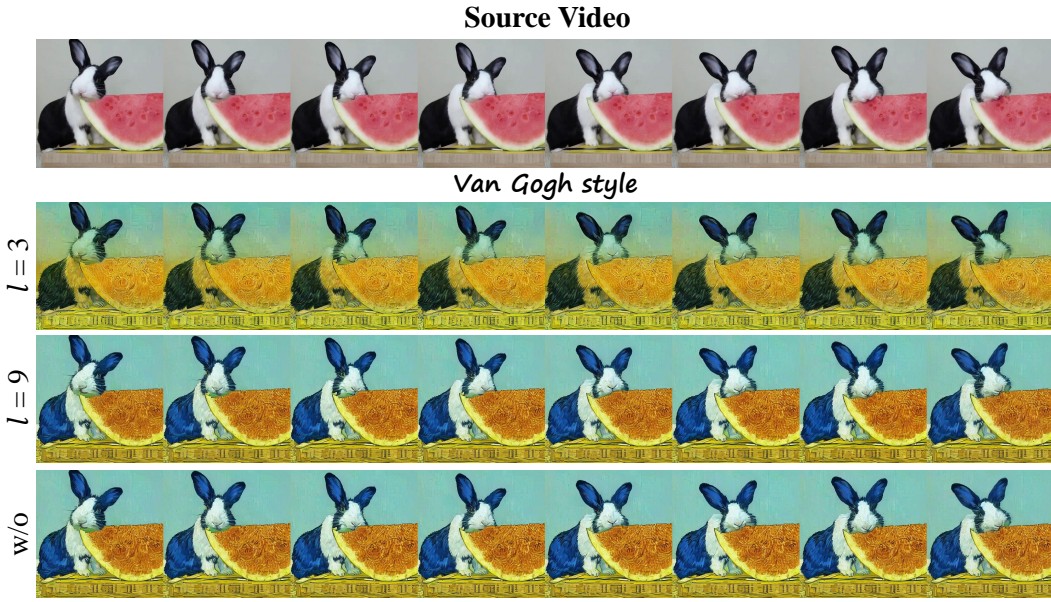

Figure 9: **Ablation Study on the window size** $l$.

## C  Visualisation of the Correspondence

We visualize the correspondence calculated by our sliding-window-based method to illustrate its effectiveness (Fig. 10). To be specific, we calculate the correspondence based on the $64 \times 64$ diffusion feature, which is extracted at the final layer of the U-net decoder. The result illustrates that our method can effectively identify the corresponding tokens.

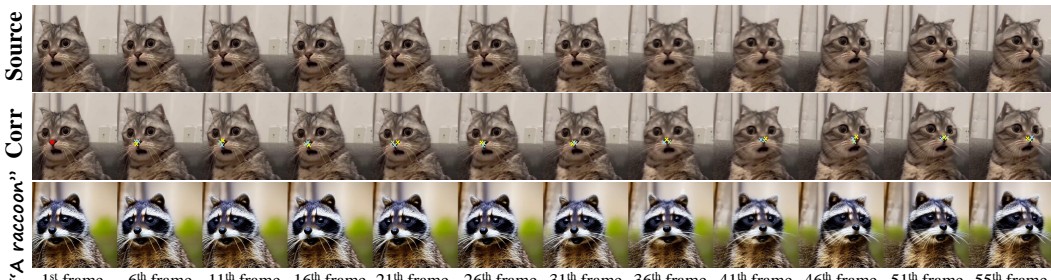

| | 1st frame | 6th frame | 11th frame | 16th frame | 21th frame | 26th frame | 31th frame | 36th frame | 41th frame | 46th frame | 51th frame | 55th frame |

Figure 10: **Visualization of the correspondence in long videos.** Given a long video, we first obtain the correspondence information ($K = 3$) through the sliding-window strategy. Then, considering a point in the first frame (the red point in the first image of the second row), we visualize the correspondence (respectively marked in yellow, green, and blue) in each frame.

## D  Accuracy of Correspondance

The correspondence acquired through the diffusion feature is accurate and robust. As there is no existing video dataset with the annotated keypoints on each frame, to further evaluate its accuracy quantitatively, we collect 5 videos with 30 frames and 5 videos with 60 frames and manually label some keypoints on each frame. Then we report the percentage of correct keypoints (PCK).

Specifically, for each video, given the first frame with the keypoints, we obtain the predicted corresponding keypoints on other frames through the diffusion feature. Then we evaluate the distance between the predicted points and the ground truth. The predicted point is considered to be correct if it lies in a small neighborhood of the ground truth. Finally, the total number of correctly predicted points divided by the total number of predicted points is the value of PCK. The result in Appendix D illustrates that the diffusion feature can accurately find the correct position in most cases for video editing.

| Method | PCK |
|---|---|
| Optical-flow Correspondence | 0.87 |
| Diffusion feature Correspondence | **0.92** |

Table 5: **Accuracy of Correspondance.**

## E  Effectiveness of correspondence guided attention during inversion

The quality of noise obtained by inversion can significantly affect the final quality of editing. The Correspondence-Guided Attention (CGA) during inversion can increase the quality and temporal consistency of the obtained noise, which can further help to enhance the quality and consistency of edited videos. The ablation of it is shown in Fig. 11

## F  Broader Impacts

Our work enables high-quality video editing, which is in high demand across various social media platforms, especially short video websites like TikTok. Using our method, people can easily create

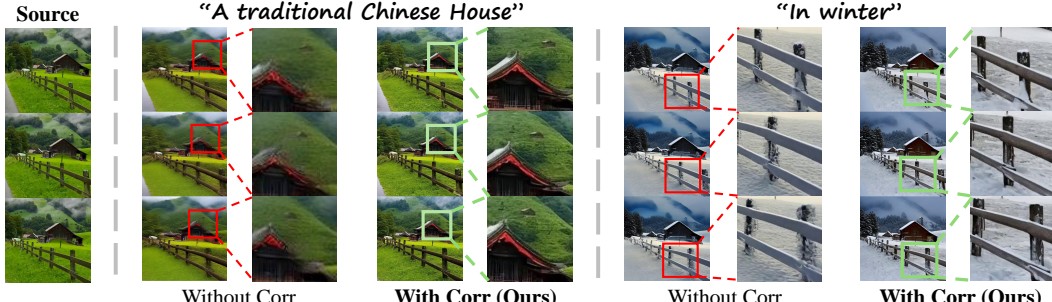

Figure 11: **Ablation Study about correspondence in inversion.** Here *Without Corr* means not applying the correspondence-guided attention during inversion, which suffers blurring and flickering. *With Corr* means the correspondence-guided attention is applied in both inversion and denoising stages, illustrating satisfying performance.

high-quality and creative videos, significantly reducing production costs. However, there is a potential for misuse, such as replacing the characters in videos with celebrities, which may infringe upon the celebrities' image rights. Therefore, it is also necessary to improve relevant laws and regulations to ensure the legal use of our method.

## G  Limitations

Despite achieving outstanding results, our methods still encounter several limitations. First, although the correspondence calculation process is efficient through the proposed sliding window strategy, the implementation of correspondence-guided attention is still not efficient enough, leading to the extra usage of GPU memory and time (Table 3). This problem is expected to be alleviated largely through the use of xFormers. We will work on it in the future.

Second, further exploration is required to optimize the application of the obtained correspondence information. In this study, we utilize the correspondence information to sample tokens during the inversion and denoising processes and do the self-attention. However, we believe that there may be more effective alternatives to self-attention that could further unleash the potential of the correspondence information.

## H  More Qualitative Results

We provide more qualitative results of our method to illustrate its effectiveness, which is shown in Fig. 13 and Fig. 12.

**Source Video**

**A red car in snowy winter**

**A Wooden car**

**A car driving in dry desert**

**A car, oil painting style**

**A car driving in sunset**

**Style of Comic Book**

**Style of Van Gogh**

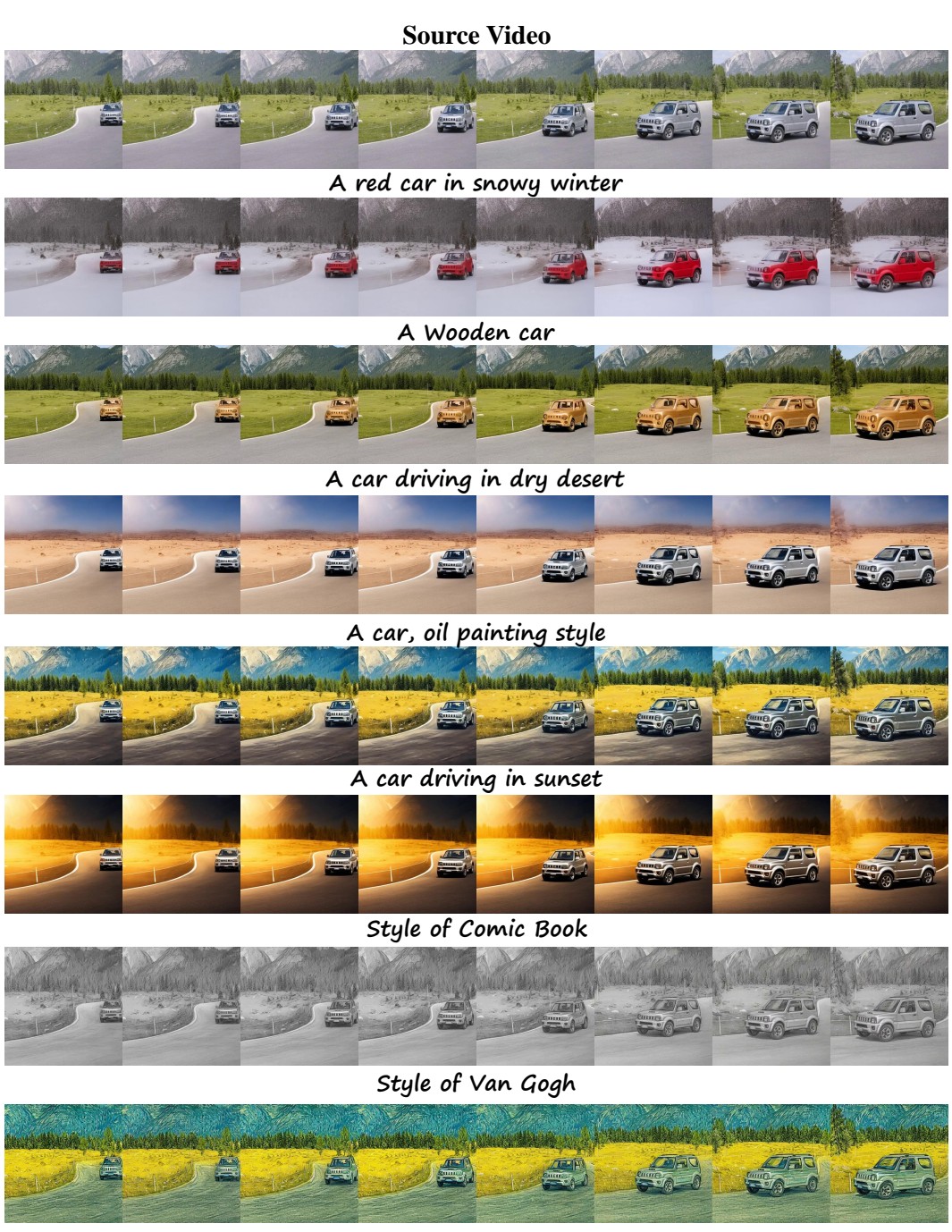

Figure 12: **Qualitative results of our methods.** Our method can effectively handle various kinds of prompts, generating high-quality videos. Results are best viewed in zoomed-in.

**Source Video**

**An African woman in grey clothes**

**A woman dressed like Black widow**

**A marble sculpture**

**Cartoon style**

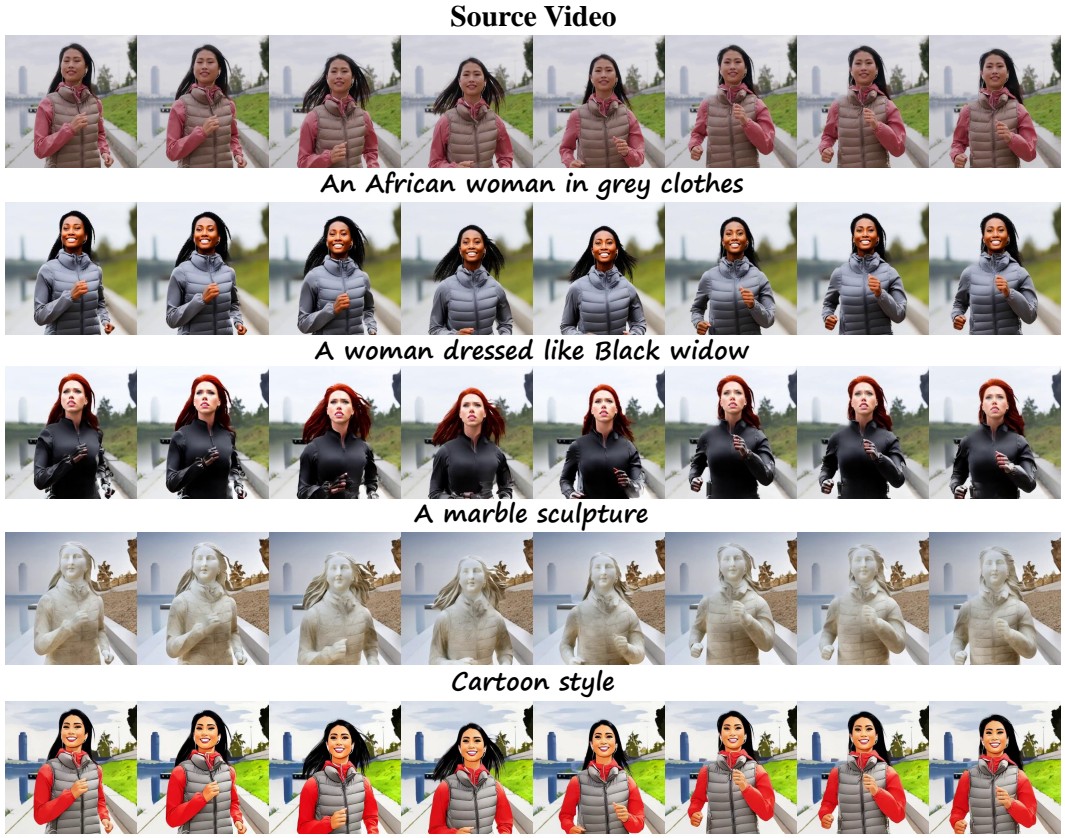

Figure 13: **Qualitative results of our methods.** Our method can effectively handle various kinds of prompts, generating high-quality videos. Results are best viewed in zoomed-in.

