# OpenReview forum: "COVE: Unleashing the Diffusion Feature Correspondence for Consistent Video Editing"
_NeurIPS.cc/2024/Conference — NeurIPS 2024 poster_

### Official Review · Reviewer_nziD · 2024-07-09

**Soundness:** 3
**Presentation:** 3
**Contribution:** 2
**Rating:** 5
**Confidence:** 5

**Summary:**

The paper introduces COrrespondence-guided Video Editing (COVE), a method to improve video editing with pretrained text-to-image (T2I) diffusion models. It addresses the challenge of maintaining temporal consistency by using diffusion feature correspondence. COVE identifies and samples highly corresponding tokens across frames, applying self-attention to them during editing. It also reduces GPU memory usage with a temporal-dimensional token merging strategy. COVE integrates seamlessly into existing T2I models without extra training and shows superior performance in various scenarios.

**Strengths:**

1. The method is reasonable and easy to follow.
2. The presentation is clear and easy to understand.
3. The Sliding-window Strategy seems useful in video editing methods.

**Weaknesses:**

1. Experimental performance improvement seems limited. Most of the editing results present a similar type involving style/color change, which is easy to implement in the previous methods from my experience. According to Fig. 6, the performance compared to earlier methods does not show high superiority, which is not convincing to support "...COVE achieves the start-of-the-art performance...".

2. I noticed that [1] also uses DIFT features to guide video editing, especially object replacement. Could you please compare COVE with it? For example, replace the "jeep" with a "sports car" or "bus" in Fig. 11. I would like to see the performance when there is a large motion/shape change.

[1] VideoSwap: Customized Video Subject Swapping with Interactive Semantic Point Correspondence

**Questions:**

How do you make sure that $k=3$ is suitable for all videos? As suggested in Fig. 2, the subject may suffer from severe morph across video frames when there is a significant motion change.

**Limitations:**

See weakness above.

---

> ### Author Rebuttal · Authors · 2024-08-07
>
> Dear Reviewer nziD,
>
> Thank you for your time and thoughtful feedback on COVE. We are pleased that you recognize the effectiveness of the sliding window strategy. We provide our feedback as follows.
> # Experimental performance
> > Experimental performance improvement seems limited. Most of the editing results present a similar type involving style/color change, which is easy to implement in the previous methods from my experience. According to Fig. 6, the performance compared to earlier methods does not show high superiority, which is not convincing to support "...COVE achieves the start-of-the-art performance...".
>
> The baseline methods mentioned in our paper are among the ***most representative and effective*** approaches in recent years, having received widespread recognition and attention. In the experiment, we present ***both quantitative and qualitative results*** to illustrate COVE's superior performance. As shown in Table 1, COVE outperforms these baseline methods on several widely used quantitative metrics. The result of user study also illustrates that edited video quality aligns with human subjective perception. For qualitative comparison, COVE successfully alleviates the problems of baselines such as blurring, temporal inconsistency, inconsistency with the prompt, etc (Figure 6). The ablation study (Table 2 and Figure 7) also illustrates that COVE can significantly enhance the temporal consistency of edited videos. We also ***provide more qualitative results*** (Figure 15 in the uploaded PDF), illustrating the ability of COVE for shape editing.
>
> # Comparasion with Baselines
> > I noticed that [1] also uses DIFT features to guide video editing, especially object replacement. Could you please compare COVE with it? For example, replace the "jeep" with a "sports car" or "bus" in Fig. 11. I would like to see the performance when there is a large motion/shape change.
>
> VideoSwap proposes to leverage the semantic point correspondence for aligning motion trajectories for high-quality video editing. However, it requires users to manually provide keypoints as the additional condition for each video, which is a relatively troublesome burden for users. Furthermore, the method needs extra training, increasing the cost of video editing.
>
> **In contrast**, COVE is a training-free framework that only requires users to provide the text prompt. Unfortunately, *the code of VideoSwap is not publicly available*, we have filled in the form provided by the authors to inquire about the source code. We are willing to compare our method with it after obtaining its source code.
>
> We also provide more experiment results as you suggested, such as changing the jeep into sports car and bus. The results are shown in Figure 15 in the uploaded PDF, illustrating the satisfying performance of our method for shape editing. We are willing to add the results to the final revision of our paper.
>
> # The value of $K$
>
> > How do you make sure that K=3 is suitable for all videos? As suggested in Fig. 2, the subject may suffer from severe morph across video frames when there is a significant motion change.
>
> As illustrated in Table 2, for the videos in our experiments, $K=3$ and $K=5$ achieve similar performance. As a result, we choose $K=3$ for a better trade-off between quality and resource consumed. What's more, $K$ is a hyperparameter in our method, which can be adjusted for each video to achieve high-quality editing.

---

> ### Author Response · Authors · 2024-08-14
>
> Dear reviewer nziD,
>
> Thanks for your valuable suggestions on our paper!
>
> We have responsed to each of your concerns in our rebuttal, including further explanation on the performance of our method, more
> qualitative results for shape editing (Figure 15 in the uploaded PDF) and the chosen value $K$.
>
> If you have any unsolved issues, please let us know. We are more than happy to answer any further questions you may have!
>
> Authors

---

### Official Review · Reviewer_Ht4d · 2024-07-13

**Soundness:** 3
**Presentation:** 3
**Contribution:** 3
**Rating:** 7
**Confidence:** 4

**Summary:**

In this paper, the authors tackle the problem of video editing using Text-to-Image diffusion models. To achieve this, the authors make use of strong diffusion model’s feature correspondence abilities. The authors propose a sliding-window based strategy to track features of source video based on correspondences. Here, the goal is to overcome the computational complexity involved with finding similarity across all patches of the video. This is achieved by computing similarity scores within a sliding window. After this, a set of corresponding tokens are collected and merged using an existing work (Token Merging paper, ICLR 2023). Experimental analysis shows that the proposed method achieves state-of-the-art performance across many tasks.

**Strengths:**

S1. Model does not require any training or even inference time optimization. The method computes correspondences in an efficient manner and track it effectively.

S2. The ideas proposed in the paper are simple and well presented overall.

S3. Results look impressive - the method is able to achieve state-of-the-art in comparison to the works in the area of video editing with T2I models.

**Weaknesses:**

W1. The paper clearly states that the base model is Stable diffusion on L208 and the model does not require any training including inflation (which I think is a major strength).  Stable diffusion model has only self-attention and cross-attention layers. It is not clear how “temporal layers” come into picture on L200. Further, it is not clear what is the 3D-Unet in context of Fig. 3.

W2. The proposed method tends to alter the regions of the image that do not correspond to the text prompt. In Fig. 1, the results shows that the proposed alters the background significantly.

W3. Results show that the proposed method is unable to alter the shape of the object, since the correspondence merging limits the shape changing ability drastically.

W4. The core novelty of exploiting the sliding window to use a lower field of comparison (as. In Eq. 10) is somewhat straightforward and limited given the obvious computational constraints. However, in conjunction with other ideas such as Token Merging [ref 1] and correspondences [ref 2], the work is novel.

W5. There have been works in the video editing space which exploit correspondences for video editing [ref 3,4]. It will be helpful to differentiate the proposed method with them. However, I would like to acknowledge that these works could have been parallel to this submission.

References

[ref 1] Token Merging: Your ViT But Faster, ICLR 2023

[ref 2] Emergent Correspondence from Image Diffusion, NeurIPS 2023

[ref 3] Space-Time Diffusion Features for Zero-Shot Text-Driven Motion Transfer, CVPR

[ref 4] GenVideo: One-shot target-image and shape aware video editing using T2I diffusion models, CVPRw

**Questions:**

- How does applying \tilde{Corr} help during inversion?

**Limitations:**

The paper addresses the limitations of the work in Appendix E.

---

> ### Author Rebuttal · Authors · 2024-08-07
>
> Dear reviewer Ht4d,
>
> Thanks for your comprehensive review and insightful comments on our paper. We appreciate that you recognize the advantages of training-free and impressive results of our method. The response to your concerns is shown below.
> # Temporal layers and 3D Unet
> > The paper clearly states that the base model is Stable diffusion $\cdots$. Further, it is not clear what is the 3D-Unet in context of Fig. 3.
>
> The temporal layer is widely applied in video diffusion models, which mainly consist of spatio-temporal attention [1]. In spatio-temporal attention, considering a query Q in one frame, the key K and value V are from both the current frame and other frames. This operation aims to capture spatio-temporal consistency of the video.
>
> To convert a 2D Unet into a 3D Unet, current methods [1] usually inflate the 2D convolution layers to pseudo-3D convolution layers, with 3x3 kernels being replaced by 1x3x3 kernels. And the self-attention in 2D Unet is replaced by the aforementioned spatio-temporal attention. We will add a detailed introduction to 3D Unet in the appendix of our paper.
>
> # Editing unrelated regions
> > The proposed method tends to alter $\cdots$ alters the background significantly.
>
>
> We would like to point out that maintaining the background unchanged is not the primary focus of our work. In fact, most current SOTA video editing methods (such as [1,2,3,4,5,6]) which ***take texts as the only condition*** also cannot maintain the background. To solve this problem, additional conditions such as a mask for regional editing need to be added, which we believe is an interesting topic for future work.
>
> On the other hand, our paper mainly focuses on maintaining temporal consistency and enhancing the quality of generated videos. The experimental results in the paper illustrate that we have effectively achieved this goal. In addition, we also notice that in some cases COVE can maintain the background unchanged (such as Figure 16 in uploaded PDF).
>
> # Shape editing
> > Results show that the proposed method is unable to alter the shape of the object, since the correspondence merging limits the shape changing ability drastically.
>
> COVE also illustrates the satisfying capability of making small modifications to the shape of the object such as changing a jeep into a sport car or a bus (Figure 15 in the uploaded PDF). We believe this is due to the correspondence can accurately reflect the motion information in the source video. This motion information remains unchanged between the source video and edited video even though the users want to make modifications to the object's shape. As a result, COVE not only significantly alleviates the problem of temporal inconsistency but also shows satisfying performance for altering the shape of objects.
>
> # Novelty
> > The core novelty of exploiting the sliding window to use $\cdots$ the work is novel.
>
> Thanks for acknowledging the novelty of our work! As stated in the paper, our proposed sliding-window method is based on the continuity of videos, which is intuitive and effective. Significantly reducing the computational complexity while achieving outstanding performance, we believe that our work is insightful for the community of video editing.
>
> # Comparasion with baselines
> > There have been works in the video $\cdots$ could have been parallel to this submission.
>
> Thanks for your advice. [6] proposes using a pretrained T2V model for motion transferring. [7] proposes target and shape-aware InvEdit masks for shape editing. However, they need extra training [7] or optimization [6]. **In contrast**, COVE is a training-free framework that can leverage popular pre-trained T2I model (such as Stable Diffusion) to achieve impressive performance. We compare our method with [6] (Figure 15 and Table 9). *The source code of [7] is not released*, we are willing to compare our method with it after the source code is released.
>
> **Table 9.** Quantitative comparison between DMT [6] and our method. We use them to respectively edit 10 20-frame videos which are widely used in the field of video editing and report the quantitative results.
> |Method|SC|MS|AQ|IQ|
> |-|-|-|-|-|
> |DMT [6]|0.9573 |0.9874 | 0.7003|0.7396 |
> |**Ours**|**0.9689**|**0.9887**|**0.7135**|**0.7447**|
>
> # Effectiveness of correspondence guided attention during inversion
> > How does applying $\tilde{Corr}$ help during inversion?
>
> The quality of noise obtained by inversion can significantly affect the final quality of editing [8, 9]. The **C**orrespondence-**G**uided **A**ttention (CGA) during inversion can increase the quality and temporal consistency of the obtained noise ($z^T$ in Figure 3), which can further help to enhance the quality and consistency of edited videos. We add the ablation study of it (Figure 16 and Table 10), which will also be added to our main paper.
>
> **Table 10.** Ablation Study of **C**orrespondence-**G**uided **A**ttention (i.e., applying $\tilde{Corr}$) during inversion.
> ||SC|MS|AQ|IQ|
> |-|-|-|-|-|
> |No CGA during inversion|0.9413|0.9740|0.7098|0.7422|
> |**CGA in both inversion and denoising**|**0.9689**|**0.9887**|**0.7135**|**0.7447**|
>
> # References
>
> [1] Tune-A-Video: One-Shot Tuning of Image Diffusion Models for Text-to-Video Generation, ICCV2023
>
> [2] FLATTEN: optical FLow-guided ATTENtion for consistent text-to-video editing, ICLR 2024
>
> [3] RAVE: Randomized Noise Shuffling for Fast and Consistent Video Editing with Diffusion Models, CVPR 2024
>
> [4] FRESCO: Spatial-Temporal Correspondence for Zero-Shot Video Translation, CVPR 2024
>
> [5] Codef: Content deformation fields for temporally consistent video processing, CVPR 2024
>
> [6] Space-Time Diffusion Features for Zero-Shot Text-Driven Motion Transfer, CVPR 2024
>
> [7] GenVideo: One-shot target-image and shape aware video editing using T2I diffusion models, CVPRw
>
> [8] Generative Rendering: Controllable 4D-Guided Video Generation with 2D Diffusion Models, CVPR 2024
>
> [9] Pix2video: Video editing using image diffusion, ICCV 2023

---

> > ### Comment · Reviewer_Ht4d · 2024-08-13
> >
> > The rebuttal addresses the concerns. I highly encourage authors to include this discussion in the paper. Hoping that authors will include these discussions in the paper, I raise my score to an "accept".

---

### Official Review · Reviewer_LETZ · 2024-07-13

**Soundness:** 3
**Presentation:** 3
**Contribution:** 3
**Rating:** 6
**Confidence:** 4

**Summary:**

This paper focused on improving the temporal consistency of video editing. They propose to leverage the inherent diffusion feature correspondence with a sliding-window based strategy. With this design, the tokens in noisy latents can be sampled based on the “one-to-many” correspondence. The experiments demonstrate superior performance than other methods.

**Strengths:**

+ The paper proposed a new diffusion feature correspondences guided video editing method. In addition, they introduce the token merging and sliding window for higher efficiency.

+ The method demonstrates superior performance than previous methods with extensive ablations to evaluate the effectiveness.

+ The paper is well-written with clear motivations.

**Weaknesses:**

- The proposed method highly relies on the correspondences of diffusion features. However, such correspondences may be difficult to obtain for videos with large content motions. The reviewer suggests a more detailed discussion about how to ensure the accuracy of the correspondences, as well as what are the potential limitations of inaccurate correspondences. In addition, it would be better to quantitatively evaluate the correctness of the correspondences of diffusion features.

- The paper mainly discussed and compared their method with optical flow-guided video editing methods. However, such correspondence-based idea is also related to deformation field based methods such as CoDeF [1] or neural atlas based methods [2], where they can ensure the accuracy of pixel / point level accuracy by evaluating the video reconstruction accuracy. The authors are suggested to compare or discuss their method with such approaches.

- The related works seem to be a bit short and less extensive. There have been many video editing works that target at improving temporal / subject consistency, efficiency, long video editing, etc. The reviewer suggests to include a more extensive related works and discuss the differences of the proposed work with them.

[1] Ouyang, Hao, et al. "Codef: Content deformation fields for temporally consistent video processing." Proceedings of the IEEE/CVF Conference on Computer Vision and Pattern Recognition. 2024.

[2] Chai, Wenhao, et al. "Stablevideo: Text-driven consistency-aware diffusion video editing." Proceedings of the IEEE/CVF International Conference on Computer Vision. 2023.

**Questions:**

o	The reviewer appreciates the qualitative examples of correspondences of diffusion features in appendix. Is there any quantitative metric to evaluate the correspondences of diffusion features?

o	How to define the temporal length for token merging? Does it remain the same for all experiment videos?

**Limitations:**

Yes.

---

> ### Author Rebuttal · Authors · 2024-08-07
>
> Dear Reviewer LETZ,
>
> Thank you for your comprehensive and detailed review of our paper and the recognition of our work's clarity and effectiveness. We provide our feedback as follows.
> # Accuracy of correspondences
> > The proposed method highly relies on the correspondences of diffusion features $\cdots$. In addition, it would be better to quantitatively evaluate the correctness of the correspondences of diffusion features.
>
> As illustrated in Figure 10 in appendix and Figure 14 in the uploaded PDF, the correspondence acquired through the diffusion feature is accurate and robust. As there is no existing video dataset with the annotated keypoints on each frame, to further evaluate its accuracy quantitatively, we collect 5 videos with 30 frames and 5 videos with 60 frames and manually label some keypoints on each frame. Then we report the percentage of correct keypoints (PCK) following the prior work [1,2] (Table 7).
>
> Specifically, for each video, given the first frame with the keypoints, we obtain the predicted corresponding keypoints on other frames through the diffusion feature. Then we evaluate the distance between the predicted points and the ground truth. The predicted point is considered to be correct if it lies in a small neighborhood of the ground truth. Finally, the total number of correctly predicted points divided by the total number of predicted points is the value of PCK. The result in Table 7 illustrates that the diffusion feature can accurately find the correct position in most cases for video editing.
>
> **Table 7.** Quantitative comparison of the accuracy between diffusion feature correspondence (DFC) and optical flow correspondence (OFC).
> |Method|PCK|
> |-|-|
> |OFC|0.87|
> |**DFC**|**0.92**|
>
> The potential limitation of inaccurate correspondences is that suboptimal results could be obtained. For instance, as discussed in Section 4.4, using a too-small window size leads to inaccurate correspondences, resulting in poor editing outcomes. By adjusting the window size, this problem can be effectively solved.
>
> # Comparasion with baselines
> > The paper mainly discussed and compared their method with optical flow-guided video editing methods $\cdots$ . The authors are suggested to compare or discuss their method with such approaches.
>
> Thanks for your advice. The deformation field-based methods [3] factorize the video into a 2D content canonical field and a 3D temporal deformation field. For each video, it requires complicated training of implicit deformable models to obtain a 2D and a 3D hash table. What's more, it still needs the optical flow model for preprocessing the video sequences.
>
> Neural atlas-based methods [4] leverage the layered representations to propagate the appearance information among frames. However, two diffusion models are employed to respectively process the background and foreground, which increases resource consumption. Furthermore, they also need extra training.
>
> **In contrast**, COVE is a training-free method that effectively obtains highly accurate correspondence simply by calculating the similarity between features. We also compare their performance (Table 8 and Figure 15 in the uploaded PDF). We are willing to add the results to the final revision of the paper.
>
> **Table 8.** Quantitative comparison among CoDeF [3], StableVideo [4] and COVE. We use them to respectively edit 10 20-frame videos which are widely used in the field of video editing and report the quantitative results.
> |Method|SC|MS|AQ|IQ|
> |-|-|-|-|-|
> |CoDeF [3]|0.9369|0.9619|0.6847|0.7134|
> |StableVideo [4]|0.9215|0.9771|0.6781|0.7273|
> |**Ours**|**0.9689**|**0.9887**|**0.7135**|**0.7447**|
>
> # Add more related works
> > The related works seem to be a bit short and less extensive $\cdots$. The reviewer suggests to include more extensive related works and discuss the differences of the proposed work with them.
>
> Thanks for your advice, we will discuss more related works in our paper, including but not limited to the CoDeF [3] and StableVideo [4].
>
> # Temporal length for token merging
> > How to define the temporal length for token merging? Does it remain the same for all experiment videos?
>
> In our experiment, the temporal length for token merging is simply set to the length of the video.
>
> # References
> [1]. Cats: Cost aggregation transformers for visual correspondence, NeurIPS 2021
>
> [2]. Transformatcher: Match-to-match attention for semantic correspondence, CVPR 2022
>
> [3]. Codef: Content deformation fields for temporally consistent video processing, CVPR 2024.
>
> [4]. Stablevideo: Text-driven consistency-aware diffusion video editing, ICCV 2023.

---

> ### Comment · Reviewer_LETZ · 2024-08-12
> **Response by Reviewer LETZ**
>
> Dear Authors,
>
> Thanks for your hard work and detailed analysis. The rebuttal has resolved all my concerns. The quantitative experiments of diffusion feature correspondences on rebuttal is helpful to demonstrate the effectiveness of leveraging diffusion features for video editing. The authors are suggested to include these results in the paper. Therefore, I decide to raise my score from 5 to 6.
>
> Best regards,
>
> Reviewer LETZ

---

### Official Review · Reviewer_sLJK · 2024-07-14

**Soundness:** 2
**Presentation:** 2
**Contribution:** 2
**Rating:** 4
**Confidence:** 4

**Summary:**

The paper proposes using the correspondence features already exist in diffusion models to find matched tokens between different frames in a video for consistent video editing. The motivation is that video editing models using optical flow to find matched features can exhibit one-to-many matching issue which limits the quality of video editing consistency when there are large motions. To overcome expensive compute, the paper proposes using "sliding window" (w.r.t frames) by only finding correspondences in subsequent frames. Reasonable qualitative and quantitative results are shown in the experiments section.

**Strengths:**

- The proposed method intuitively make sense
- The flow of the paper is relatively easy to follow
- The results shown are reasonable

**Weaknesses:**

- Optical flow is also essentially just finding matched patches/features between different frames. The proposed way of finding one-to-many correspondences between frames can also be applied uusing optical flow (with some small modifications of existing optical flow algorithms). I do not see any evidence from the paper that using the features in a diffusion model is better than using the features extracted by an optical flow algorithm (explicitly or inexplicitly depending on different optical flow methods)

- the above should be an important ablation study that is currently missing from the paper

- the sliding window algorithm, though simple and achieving compute reduction, can also lead to errors accumulating across frames when editing long videos

- the novelty is a bit limited. My argument is the following:
    1. Features inexplicitly extracted in Stable Diffusion can be used for feature correspondence is well-known to the community, e.g. in Tang et al. NeurIPS 2023
    2. Finding correspondences for video editing is widely studied and shown to be effective, including but not limited to the several references this paper already cited
    3. The main contribution then seems to be proposing using SD's features to find correspondences instead of using optical flow. However, the one-to-many problem and many-to-one problem is essentially a thresholding problem when finding correspondences and the receptive field size used to find matched patches, and I cannot see why the features extracted by SD is superior than the features extracted by a state-of-the-art optical flow model. This ablation study is missing

**Questions:**

See weakness

---

> ### Author Rebuttal · Authors · 2024-08-07
>
> Dear Reviewer sLJK,
>
> Thanks for your time and thoughtful review. We appreciate your recognition of the satisfying experimental results and clear writing of the paper. We provide our feedback as follows.
> # Optical flow model for one-to-many correspondences
> > Optical flow is $\cdots$ by an optical flow algorithm.
>
> Thanks for your valuable suggestions. Modifying the optical flow model [1,2] from one-to-one to one-to-many correspondence may seem simple and intuitive. Unfortunately, we would like to point out that it is still nontrivial in practice due to the inherent characteristics of optical flow models.
> - For the $i$th frame $I_i$ (with the shape of [H,W]) in a video, the optical flow model ***directly predicts*** an optical field $F$ with the shape of [H,W,2], representing the offset of each pixel along $x$ and $y$ axis in $I_{i+1}$. The prediction of $F$ is a ***regression problem*** during training, where L1 loss is applied between $F$ and the ground truth. In other words, for each pixel, optical flow models ***directly predict*** the position of the ***single*** corresponding pixel in the next frame, rather than firstly calculate confidence for each pixel in the next frame and then select the pixel with the highest confidence.
> - Based on the analysis above, the off-the-shelf optical flow model cannot obtain the one-to-many correspondence. Furthermore, there is also ***no existing dataset with the annotations of one-to-many correspondence*** to support the training of a model with modified structures from scratch . The construction of dataset and the design of new model structure are still unexplored by existing works, which is out of the scope of our paper.
>
> As a result, current optical flow models fail to obtain one-to-many correspondence. Instead, we provide the ablation study between COVE with $K=1$ (i.e., one-to-one correspondence) and flow-based baseline (Figure 13 and Table 5). The results illustrate that even with one-to-one correspondence, COVE still achieves comparable or even better result compared with the flow-based baseline. Furthermore, one of the main advantages of COVE is that it can effectively obtain one-to-many correspondence. With the increase of $K$, the quality of edited videos is better, achieving SOTA performance. We will update the details in the paper.
>
> **Table 5.** Quantitative comparison between OFC(optical-flow correspondence) and DFC(diffusion feature correspondence). The detailed experiment settings follow original paper.
> |Method|SC|MS|IQ|AQ|
> |-|-|-|-|-|
> |OFC|0.9617|0.9622|0.7155|0.6544|
> |**DFC ($K=1$)**|0.9637|0.9817|0.7148|0.6979|
> |**DFC ($K=3$)**|**0.9731**|**0.9892**|**0.7441**|**0.7122**|
> # Accuracy of the sliding window
> > The sliding window $\cdots$ long videos.
>
> Thanks for your advice. In our method, the position of the sliding window is dynamically adjusted rather than fixed in each frame (Figure 4), which can ensure the accuracy of correspondence in long video. We also add the experiment about the editing quality of 10 short videos (20 frames) and their longer counterpart (60 frames). The results (Table 6 and Figure 14) illustrate that COVE is also effective for longer videos. We also provide the quantitative results about accuracy of correspondence in Table 7 (in feedback for Reviewer LETZ).
>
> **Table 6.** Quantitative comparison between the editing quality of 20 frames and 60 frames.
> |Video length|SC|MS|IQ|AQ|
> |-|-|-|-|-|
> |20|0.9689|0.9887|0.7447|0.7135|
> |60|0.9685|0.9882|0.7441|0.7140|
>
> What's more, the window size is a hyper-parameter. We may enlarge the window size if there are extremely sharp and fast motions (although such videos are very rare in real world). Even with a larger window size, our method still greatly reduces complexity.
> # Novelty
> We believe COVE elegantly addresses several challenging limitations in previous SOTA methods, potentially offering valuable insights for further research.
> > 1.Features inexplicitly $\cdots$ to the community.
>
> Existing work has primarily focused on correspondence between two images. However, it remains to be explored whether this correspondence can maintain consistent stability and accuracy over multiple frames and longer time scales in videos. Furthermore, whether the correspondence can aid in video editing tasks also lacks exploration. COVE elegantly proposes using correspondence to guide attention during inversion and denoising procedures. Achieving outstanding results (Figure 15 in the uploaded PDF), our work illustrates that ***using diffusion features to enhance quality and consistency*** is an insightful topic in the field of video editing.
>
> > 2.Finding correspondences $\cdots$ be effective.
>
> Although some previous methods also explore the correspondence for video editing, they heavily depend on the availability of a pretrained optical flow model with high accuracy, which is often not feasible in practice. Until now, acquiring correspondence information without optical flow models is still a challenging problem for video editing. Addressing this, COVE successfully illustrates that the ***optical flow model is not indispensable for finding correspondence in video editing***, and achieves ***even better*** results compared to flow-based methods.
>
> > 3.The main contribution $\cdots$ is missing.
>
> As stated above, in the optical flow model, the position of corresponding token is directly predicted through regression rather than firstly calculate the similarity in the feature and then select the token with highest similarity. As a result, the one-to-many and one-to-one problem ***is not a simple thresholding problem in optical flow models***. COVE elegantly addresses this limitation, obtaining one-to-many correspondence simply and effectively. So, we believe that our work is an insightful and interesting contribution to the community.
> # References
> [1] RAFT: Recurrent All-Pairs Field Transforms for Optical Flow, ECCV 2020 Best Paper
>
> [2] GMFlow: Learning optical flow via global matching, CVPR 2022

---

> ### Author Response · Authors · 2024-08-14
>
> Dear reviewer sLJK,
>
> Thanks for your valuable advice on our paper.
>
> We have responsed to each of your concerns in our rebuttal, including the comparasion between optical-flow correspondence and diffusion feature correspondence (Table 5, Table 7 and ***Figure 13 in the uploaded PDF***), accuracy of sliding window method on long videos (Table 6 and ***Figure 14 in the uploaded PDF***), and the novelty of our method. We also point out that using the optical flow model to obtain the one-to-many correspondence is nontrivial.
>
> If you have any unsolved questions, please let us know. We are more than happy to answer any further questions you may have!
>
> Authors

---

### Author Rebuttal · Authors · 2024-08-07

# Overall reply for all reviewers

We thank all reviewers for their constructive comments. We have responded to each of the concerns below. In the following response, Figure 1-12 and Table 1-4 are in our main paper, ***Figure 13-16 are in the uploaded PDF of the rebuttal, and Table 5-10 are in our response***. For quantitative comparison of the video quality, we still report the four popular metrics used in the paper, i.e., Subject Consistency (SC), Motion Smoothness (MS), Aesthetic Quality (AQ), and Imaging Quality (IQ).

---

> ### Author Response · Authors · 2024-08-12
>
> Dear all reviewers,
>
> Thanks for your time and effort in reviewing our paper!
>
> We are pleased that you acknowledge the strength of our paper such as its clear motivation and superior performance. **We have responsed to each of your concerns in our rebuttal. If you have any unresolved issues, please let us know.** We are more than happy to answer any further questions you may have!
>
> Thank you once again for your consideration.

---

### Decision · Program_Chairs · 2024-09-25

**Decision:**

Accept (poster)

**Comment:**

The paper received ratings of $(7, 6, 5, 4)$ after the rebuttal and discussions. **Reviewer Ht4d** and **Reviewer LETZ**, who gave ratings of $7$ and $6$, acknowledged that the authors' responses addressed their concerns and thus expressed their support. The remaining two reviewers, while having borderline opinions, did not raise further issues after the rebuttal. Specifically, the rebuttal provided clarifications with additional figures and tables to address **Reviewer sLJK**'s questions about the differences with optical flow models, the robustness of using sliding windows, and the novelty of the work.

In light of these considerations, the area chair recommends accepting this paper. The authors are encouraged to incorporate the discussions and additional figures and tables in the final version. Furthermore, it is imperative that the authors address the ethical concerns raised by the Ethics Reviewers regarding potential misuse and the user study involving human subjects.